# Model-Based Design of Induction Motor Control System in MATLAB

**Tibor Krenicky** [1,*] , **Yury Nikitin** [2] **and Pavol Božek** [3]

1 Department of Technical Systems Design and Monitoring, Faculty of Manufacturing Technologies with a seat in Prešov, Technical University of Košice, 080 01 Prešov, Slovakia
2 Department of Mechatronic Systems, Kalashnikov Izhevsk State Technical University, Izhevsk 426069, Russia
3 Institute of Production Technologies, Faculty of Materials Science and Technology in Trnava, Slovak University of Technology in Bratislava, 917 24 Trnava, Slovakia
* Correspondence: tibor.krenicky@tuke.sk

**Abstract:** A mathematical model of induction motor (IM) based on the second Kirchhoff's law with Maxwell's equation taken into account has been developed. A mathematical model of a three-phase induction motor with a short-circuited rotor in phase axes without taking into account the saturation of the magnetic circuit and the losses in steel has been elaborated. A nonlinear model of asynchronous motor in the state space in the rotating system (*d-q*) of coordinates synchronized with the stator flux and a simulation model of IM in MatLab/Simulink software, with the possibility of setting angular velocity of rotation and resistance torque, have also been developed. The model-oriented design of the control program is performed on the example of digital signal processors from Texas Instruments on the LAUNCHXL-F28379D board. This description of the dynamics provides a direct insight into the physical processes in IMs. The advantage of the considered mathematical description of electromechanical energy conversion processes in three-phase IM is that it uses instantaneous values of currents and voltages of stator and rotor winding phases as variables.

**Keywords:** induction motor; nonlinear dynamic model; MatLab/Simulink; LAUNCHXL-F28379D



## 1. Introduction

The induction motor (IM) is widely used in industry due to its reliability and low cost. Today, IMs have found their application in various industries. IMs have a high efficiency factor—80–96%, as well as a minimal price/reliability ratio. IMs with squirrel-cage winding are actively used in various industries, their share making up about 70% of all types of electric motors. The power of AC motors required in production facilities can range from several kilowatts to megawatts [1].

This paper improves the vector control system for IM based on the model approach. For this purpose, a mathematical model of the motor in the form of Cauchy equations is developed. Due to the vector control, the motor has good dynamic performance, and it can quickly accelerate and brake according to the set speed. Another problem, which is solved in the article, is the development of the control system software based on a model-based design. This approach dramatically reduces the time required to prepare control programs. The control program is formed on the basis of the model algorithm, and the number of programming errors is significantly reduced. Thus, mathematical and simulation methods and model-based designs are used in this work. Ultimately, the application of these methods reduces the time of the development of control programs, improving the quality of motor control.

The IMs principle of operation provides a numerous advantage in comparison with other types of motors, including:

(1)  Ease of fabrication;
(2)  Relatively low cost;
(3)  Low power consumption;
(4)  Versatility of application;
(5)  High reliability in operation;
(6)  Possibility of connection into the network without power converters.

The main disadvantages of IMs are the following:

(1)  Significant power losses for heat, the possibility of overheating the motor;
(2)  Lack of stability in maintaining the set speed;
(3)  Low starting torque;
(4)  Considerable starting current;
(5)  Low power factor.

A significant number of the disadvantages can be eliminated by feeding the IM from a static frequency converter. The main point is that the equipment that uses this type of motor must not have precise positioning [2]. For example, IMs are used in automated equipment drives of rolling and drawing machines in the metallurgical industry. IMs are used in all machine tools, machining centers, conveyors, crane equipment, centrifuges, industrial washing machines, etc. in the metalworking industry. IMs are used in mechanical assembly production: in drives of manipulators, conveyors, and compressors.

IMs are used in mining: in drilling and excavator equipment. IMs are found in the domestic sphere of human activity: in hand-held electric tools, washing machines, pumps, extractors, fans, and other equipment [3].

Control in the coordinate system oriented by the calculated rotor flux vector ($\psi_R$) is called direct-field oriented control. Usually, systems with direct orientation also contain a flow stabilization loop. However, a simpler approach and another class of systems has emerged in which the rotor flux vector ($\psi_R$) is not evaluated, but its angle is calculated from the slip and velocity (or position) of the rotor. Such a system is called a system with indirect field orientation [4].

Such system do not contain a flow stabilization loop. In this case, it is believed that if the control actions on the motor are formed based on the condition of flow constancy, then if they are accurately implemented, the flow will be constant even without stabilization [5].

As practice has shown, this approach has justified itself due to a good ratio of accuracy and simplicity of regulation. A promising direction in the theory and practice of motor control is vector control. Its mathematical basis is differential equations describing IM equally correctly both in dynamics and in statics. Torque is obtained by controlling the amplitude and instantaneous phase of stator current vector or stator voltage vector. Vector control, contrary to scalar control, allows an assembly of highly dynamic and precise IMs, providing the highest speed and control precision [6].

There are many tasks where it is necessary to provide a given speed, and the described disadvantage becomes very relevant. In such cases, vector frequency control is used, in which the controller calculates the voltage needed to maintain torque, providing a stable frequency. In contrast to scalar mode, there is a "smart" control of rotor magnetic flux [7]. Vector control of IM is especially relevant at low frequencies—below 10 Hz, when the operating torque of the motor is dropping significantly. In addition, this method allows to keep stable speed (with predictable linear change) during acceleration. This is achieved by obtaining a high starting torque until the motor reaches steady mode [8].

It is also important that vector control saves energy (in some cases up to 60%), because most of the time the frequency converter sends exactly as much energy to the motor as necessary to maintain the set speed [9].

At present, the most perspective method of IM control is frequency regulation, which became possible due to the intensive development of power electronics element base and computer technology. Nowadays almost all controlled electric drives based on IMs are controlled by frequency converters (FC) [10].

FC manufacturers use standard control algorithms, such as scalar (with various types of correction), current vector control, voltage vector control, and both open (sensorless) systems and closed systems with speed loop, flow loop, etc. Furthermore, Siemens and ABB companies use direct torque control (DTC) [11].

More complex algorithms can be used, adapted to the specific applications in small series and specialised FCs. Complex observers are developed to measure a number of parameters required to ensure their operability, introducing additional errors into the sensorless control systems. There is a practice of construction of multiloop control systems, significantly improving the characteristics of IMs. However, it also has a number of problems, for example, building dual-circuit systems with a current loop requires additional sensors to be installed on the motor, or, also, building quite complex observers. The construction of multi-circuit systems with serial FCs is not always possible due to the lack of appropriate functions in the software of such frequency converters and the inability to change their software [12].

Specialized and small-scale FCs have a high price, and are not universal (most often they are designed for use only on specific objects). FCs requires additional sensors to be installed on the motor or on the process unit. These disadvantages nullify the main advantages of using asynchronous motors—low price, reliability, low maintenance costs [13].

Vector control represents a method for how to control a variable frequency drive where the stator currents of a three-phase IM are conceptualized in the form of two orthogonal components that can be visualized as vectors, where the first component identifies the magnetic flux of the motor, while the other component identifies the torque. The IM control system calculates the appropriate current component assignments based on the magnetic flux and torque assignments specified by the system controlling the drive speed. In most cases, the usage of the proportional-integral (PI) controllers is to maintain the measured components of current on their reference values. In this way, the pulse-width modulation of the variable frequency drive controls the transistors switching as determined by the given stator voltages, which are the outputs of the PI current controllers [14].

Vector control is suitable for controlling synchronous or IMs [15]. Initially, it was established for high-performance motors which are required to operate smoothly over the whole range of speeds and provide full torque also at zero speed. That also includes high dynamic performance such as rapid acceleration and deceleration. Thus, vector control is increasingly demanded in various fields of the industry, as IMs provide important advantages in comparison with DC motors. Thanks to the increasing computing performance of the control electronics, vector control is expected to eventually supplant scalar control in most applications [16].

Vector control allows that IMs or synchronous motors can be driven for all operating conditions, similar to DC motors [17]. That means the AC motor functions in the same way as a DC motor where the magnetic field coupling and the armature flux coupling produced by the corresponding field and armature current (or torque component) are orthogonally directed. When torque is controlled, the flux coupling is not affected by the field, and thus the dynamic torque response is provided [18].

Vector control respectively generates the output voltage of the three-phase PWM motor originating from the complex voltage vector utilised to operate the complex current vector that is obtained from the input three-phase motor stator current by utilizing the projections or backward and forward rotations between the three-phase system, dependent on the time and speed into a two-coordinate time-dependent rotating reference system appropriate for these vectors description [19].

The complex spatial stator current vector is defined in the coordinate system (d, q) where orthogonal components belong to the straight (d) and quadrature (q) axes. In that way, the magnetic coupling component of the current field is assigned to the d axis while the current torque component is assigned to the q axis [20]. Moreover, the described system of coordinates (d, q) for the induction motors is superimposed on the three-phase sinusoidal system of the instantaneous motor (a, b, c). The components (d, q) of the system's current

vector permit standard control regimes including integral, proportional, or PI control, similarly to a DC motor case [21].

The (d, q) coordinate system projections generally include:

- The direct projection of instantaneous currents to a complex spatial vector representation (a, b, c) of stator currents, using a three-phase sinusoidal system;
- The direct two-phase projection of (a, b, c) onto (α,) utilising the Clark transform. For the implementation of vector control, a motor with balanced three-phase currents is standardly assumed providing those two phases of the motor current are to be measured only. In addition, the reverse projection from (α,) to (a, b, c) utilises an inverse Clark transform or PWM modulator space-vector along with one of the diverse PWM modulators [22].

The Clark transform is utilized in order to convert a system of three-phase currents and voltages into a two-coordinate time-independent linear system. This makes it possible to utilise uncomplicated and easy-to-implement PI controllers, as well as to simplify the control of currents that generate magnetic flux and torque [23].

The rotation speed of the (d, q) coordinate system may be adjusted to any intended value. However, there could be preferred following speeds of the reference system:

- Non-rotating (d, q) coordinate system within the stationary reference frame;
- The (d, q) coordinate system rotating at a synchronous speed within the reference system;
- The coordinate system (d, q) rotates at a rotor speed within the rotor reference system.

In order to find an effective control algorithm, the torque and excitation currents may be inferred using the original stator current inputs.

In DC motors, the magnetic field and torque components may be relatively simply controlled using the separate controlling of the corresponding field and armature currents. The economic control of variable speed AC motors requires microprocessor control with AC drives using a high-performance digital signal processing (DSP) technology [24].

Inverters have good potential for the sensorless or feedback-including implementation. Their key limitation in the case of a non-feedback operation is the minimum speed possible at full torque [25].

Two methods of vector control are serviceable: direct field oriented control (DFOC) and sensor (indirect) field oriented control (IFOC) that is used more often, as these drives are easier to operate over the entire speed ranging from zero to high speeds in closed loop mode.

In the case of feedback vector control, feedback signals that represent angle and magnitude are calculated directly with use of current or voltage model. In the case of direct-coupled control, the flux magnitude feedback and space angle signals in essence represent the stator current and rotor speed. Consequently, by summing the rotor angle they obtain their intrinsic flux space angle that corresponds to the rotor speed and the determined slip angle reference value corresponding to the slip frequency [26].

IMs sensorless control is desirable in terms of reliability and price. Pre-requisitions for the sensorless control comprise acquiring information about rotor speed from the measured stator electric current and voltage that is combined with open-loop estimates or closed-loop observers [27]. The software tools for the nonlinear control systems development has enabled many opportunities for ongoing progress of control systems design. Amongst them, MATLAB represents one of the most popular software platforms for analysis of linear and nonlinear control systems, with the techniques used in nonlinear control dynamical systems such as feedback linearization technique along with sliding mode control using the controller parameters [28].

The main aim of the presented article is the description of a newly developed mathematical simulation model of the induction motor in the rotating coordinate system in the state space, allowing to set the angular velocity of rotation and moment of resistance.

## 2. Materials and Methods

The rotor coordinate system is rotating relatively to the coordinate system of the stator with angular velocity $\omega$, their mutual arrangement is characterized by the electric angle $\alpha$ between the same-name axes.

The initial system of differential equations of three-phase asynchronous electric motor can be written in vector-matrix form Equation (1) based on the second Kirchhoff's law with regard to Maxwell's equation:

$$[u] = [R][i] + \frac{d}{dt}[\Psi] = [R][i] + \frac{d}{dt}([L][i]) = [R][i] + [L]\left(\frac{d}{dt}[i]\right) + \left(\frac{d}{dt}[L]\right)[i], \quad (1)$$

where $[R]$ is the matrix of active resistances; $[u]$ is the voltage vector; $[\Psi]$ is the flux vector; $[i]$ is the vector of electric currents; and $[L]$ is the matrix of inductances.

Matrices and vectors of three-phase IM enter the Equation (1), and have the form Equations (2)–(4):

$$[u] = \begin{bmatrix} [u_1] \\ [u_2] \end{bmatrix} = \begin{bmatrix} u_{1A} \\ u_{1B} \\ u_{1C} \\ u_{2A} \\ u_{2B} \\ u_{2C} \end{bmatrix}; [i] = \begin{bmatrix} [i_1] \\ [i_2] \end{bmatrix} = \begin{bmatrix} i_{1A} \\ i_{1B} \\ i_{1C} \\ i_{2A} \\ i_{2B} \\ i_{2C} \end{bmatrix}; [\Psi] = \begin{bmatrix} [\Psi_1] \\ [\Psi_2] \end{bmatrix} = \begin{bmatrix} \Psi_{1A} \\ \Psi_{1B} \\ \Psi_{1C} \\ \Psi_{2A} \\ \Psi_{2B} \\ \Psi_{2C} \end{bmatrix}, \quad (2)$$

$$[R] = \begin{bmatrix} [R_1] & [0] \\ [0] & [R_2] \end{bmatrix} = \begin{bmatrix} R_{1o} & 0 & 0 & 0 & 0 & 0 \\ 0 & R_{1o} & 0 & 0 & 0 & 0 \\ 0 & 0 & R_{1o} & 0 & 0 & 0 \\ 0 & 0 & 0 & R_{2o} & 0 & 0 \\ 0 & 0 & 0 & 0 & R_{2o} & 0 \\ 0 & 0 & 0 & 0 & 0 & R_{2o} \end{bmatrix}; \quad (3)$$

$$[L] = \begin{bmatrix} [L_1] & [M_{12}] \\ [M_{21}] & [L_2] \end{bmatrix} = \begin{bmatrix} L_{1o} & M_{1A1B} & M_{1A1C} & M_{1A2A} & M_{1A2B} & M_{1A2C} \\ M_{1B1A} & L_{1o} & M_{1B1C} & M_{1B2C} & M_{1B2B} & M_{1B2C} \\ M_{1C1A} & M_{1C1B} & L_{1o} & M_{1C2A} & M_{1C2B} & M_{1C2C} \\ M_{2A1A} & M_{2A1B} & M_{2A1C} & L_{2o} & M_{2A2B} & M_{2A2C} \\ M_{2B1A} & M_{2B1B} & M_{2B1C} & M_{2B2A} & L_{2o} & M_{2B2C} \\ M_{2C1A} & M_{2C1B} & M_{2C1C} & M_{2C2A} & M_{2C2B} & L_{2o} \end{bmatrix}, \quad (4)$$

where $[i_1]$, $[\Psi_1]$, $[u_1]$ are vectors of stator electric currents, stator flux vector and voltages of stator winding phases; $[L_1]$, $[R_1]$ are matrices of active inductances and stator winding resistances; $[i_2]$, $[\Psi_2]$, $[u_2]$ are vectors of rotor electric currents, rotor flux vector and voltages of rotor winding phases; $[L_2]$, $[R_2]$ are matrices of inductances and active resistances of the rotor winding, reduced to the stator winding; $[0]$ is zero matrix; $[M_{21}]$ is matrix of mutual inductances of the rotor and stator; $[M_{12}]$ is matrix of mutual inductances of stator and rotor.

The indices 1A, 1B, 1C indicate that the parameter belongs to the corresponding stator phase and the indices 2A, 2B, 2C indicate that the parameter belongs to the corresponding rotor phase. The index *o* indicates that the parameter belongs to a three-phase asynchronous electric machine with the parameters of the rotor winding connected to the stator.

The differentiation of the matrix $[L]$, is written in the following form (5):

$$\frac{d[L]}{dt} = \frac{d\alpha}{dt}\frac{\partial[L]}{\partial\alpha} = \omega\frac{\partial[L]}{\partial\alpha}. \quad (5)$$

Taking into account the adopted notations, system (5) is written in the form (6):

$$\begin{bmatrix} [u_1] \\ [u_2] \end{bmatrix} = \begin{bmatrix} [R_1] & [0] \\ [0] & [R_2] \end{bmatrix} \begin{bmatrix} [i_1] \\ [i_2] \end{bmatrix} + \begin{bmatrix} [L_1] & [M_{12}] \\ [M_{21}] & [L_2] \end{bmatrix} \frac{d}{dt}\left(\begin{bmatrix} [i_1] \\ [i_2] \end{bmatrix}\right) + \frac{d}{dt}\left(\begin{bmatrix} [L_1] & [M_{12}] \\ [M_{21}] & [L_2] \end{bmatrix}\right)\begin{bmatrix} [i_1] \\ [i_2] \end{bmatrix}, \quad (6)$$

The stator and rotor inductance matrices included in Equation (6) are written as Equations (7) and (8):

$$[L_1] = \begin{bmatrix} L_{1o} & M_{1A1B} & M_{1A1C} \\ M_{1B1A} & L_{1o} & M_{1B1C} \\ M_{1C1A} & M_{1C1B} & L_{1o} \end{bmatrix} = \begin{bmatrix} L_{1o} & M_m \cos \frac{2\pi}{3} & M_m \cos \frac{4\pi}{3} \\ M_m \cos \frac{4\pi}{3} & L_{1o} & M_m \cos \frac{2\pi}{3} \\ M_m \cos \frac{2\pi}{3} & M_m \cos \frac{4\pi}{3} & L_{1o} \end{bmatrix} = \begin{bmatrix} L_{1o} & -\frac{M_m}{2} & -\frac{M_m}{2} \\ -\frac{M_m}{2} & L_{1o} & -\frac{M_m}{2} \\ -\frac{M_m}{2} & -\frac{M_m}{2} & L_{1o} \end{bmatrix}, \tag{7}$$

$$[L_2] = \begin{bmatrix} L_{2o} & M_{2A2B} & M_{2A2C} \\ M_{2B2A} & L_{2o} & M_{2B2C} \\ M_{2C2A} & M_{2C2B} & L_{2o} \end{bmatrix} = \begin{bmatrix} L_{2o} & M_m \cos \frac{2\pi}{3} & M_m \cos \frac{4\pi}{3} \\ M_m \cos \frac{4\pi}{3} & L_{2o} & M_m \cos \frac{2\pi}{3} \\ M_m \cos \frac{2\pi}{3} & M_m \cos \frac{4\pi}{3} & L_{2o} \end{bmatrix} = \begin{bmatrix} L_{2o} & -\frac{M_m}{2} & -\frac{M_m}{2} \\ -\frac{M_m}{2} & L_{2o} & -\frac{M_m}{2} \\ -\frac{M_m}{2} & -\frac{M_m}{2} & L_{2o} \end{bmatrix}, \tag{8}$$

where $M_m$ is the maximum value of mutual inductance arising from the alignment of phases.

The assumption made earlier about the uniformity of the air gap allows us to conclude that all the eigeninductances of phases do not depend on the angular position of the rotor and are determined by Equation (9).

$$L_{1o} = L_{1co} + L_{1\sigma0}; \; L_{2o} = L_{2co} + L_{2\sigma0}; \tag{9}$$

where $L_{1co}$ is stator phase inductance from the main magnetic flux;

$L_{1\sigma0}$ is inductance from stator magnetic flux dissipation;

$L_{2co}$ is the inductance of the rotor phase reduced to the stator from the main magnetic flux;

$L_{2\sigma0}$ is inductance reduced to the stator from the magnetic flux of the rotor dissipation.

For a given IM, the maximum value of mutual inductance $M_m$ is equal to the inductance from the main magnetic flux of the phase and is related to the inductance of the magnetizing circuit by equality (10):

$$L_{1co} = L_{2co} = M_m = \frac{2}{3} L_o \tag{10}$$

Let us write the matrices of mutual inductances as Equations (11) and (12):

$$[M_{12}] = \begin{bmatrix} M_{1A2A} & M_{1A2B} & M_{1A2C} \\ M_{1B2A} & M_{1B2B} & M_{1B2C} \\ M_{1C2A} & M_{1C2B} & M_{1C2C} \end{bmatrix} = M_m \times \begin{bmatrix} \cos \alpha & \cos\left(\alpha + \frac{2\pi}{3}\right) & \cos\left(\alpha + \frac{4\pi}{3}\right) \\ \cos\left(\alpha + \frac{4\pi}{3}\right) & \cos \alpha & \cos\left(\alpha + \frac{2\pi}{3}\right) \\ \cos\left(\alpha + \frac{2\pi}{3}\right) & \cos\left(\alpha + \frac{4\pi}{3}\right) & \cos \alpha \end{bmatrix}; \tag{11}$$

$$[M_{21}] = \begin{bmatrix} M_{2A1A} & M_{2A1B} & M_{2A1C} \\ M_{2B1A} & M_{2B1B} & M_{2B1C} \\ M_{2C1A} & M_{2C1B} & M_{2C1C} \end{bmatrix} = [M_{12}]^T = M_m \times \begin{bmatrix} \cos \alpha & \cos\left(\alpha + \frac{4\pi}{3}\right) & \cos\left(\alpha + \frac{2\pi}{3}\right) \\ \cos\left(\alpha + \frac{2\pi}{3}\right) & \cos \alpha & \cos\left(\alpha + \frac{4\pi}{3}\right) \\ \cos\left(\alpha + \frac{4\pi}{3}\right) & \cos\left(\alpha + \frac{2\pi}{3}\right) & \cos \alpha \end{bmatrix}, \tag{12}$$

where T is the transpose sign of the matrix.

From Equation (13), it is possible to determine the electric angle of rotation of the rotor:

$$\alpha(t) = \alpha(0) + \int_0^t \omega(t) dt, \tag{13}$$

where $\omega$ is the angular velocity of the rotor, el. rad/s.

Since the elements of matrices $[L_1]$ and $[L_2]$ according to Equations (7) and (8) are independent on the angular position of the rotor, by taking partial derivatives of matrix elements by rotor rotation angle $\alpha$, we obtain zero matrices of size 3 × 3.

Taking partial derivatives of the elements of matrices $[M_{12}]$ and $[M_{21}]$ on the rotor rotation angle $\alpha$, we obtain Equations (14) and (15):

$$\frac{\partial[M_{12}]}{\partial\alpha} = -M_m \times \begin{bmatrix} \sin\alpha & \sin\alpha\left(\alpha + \frac{2\pi}{3}\right) & \sin\alpha\left(\alpha + \frac{4\pi}{3}\right) \\ \sin\alpha\left(\alpha + \frac{4\pi}{3}\right) & \sin\alpha & \sin\alpha\left(\alpha + \frac{2\pi}{3}\right) \\ \sin\alpha\left(\alpha + \frac{2\pi}{3}\right) & \sin\alpha\left(\alpha + \frac{4\pi}{3}\right) & \sin\alpha \end{bmatrix}; \qquad (14)$$

$$\frac{\partial[M_{21}]}{\partial\alpha} = -M_m \times \begin{bmatrix} \sin\alpha & \sin\alpha\left(\alpha + \frac{4\pi}{3}\right) & \sin\alpha\left(\alpha + \frac{2\pi}{3}\right) \\ \sin\alpha\left(\alpha + \frac{2\pi}{3}\right) & \sin\alpha & \sin\alpha\left(\alpha + \frac{4\pi}{3}\right) \\ \sin\alpha\left(\alpha + \frac{4\pi}{3}\right) & \sin\alpha\left(\alpha + \frac{2\pi}{3}\right) & \sin\alpha \end{bmatrix}. \qquad (15)$$

By multiplying the vectors and matrices in Equation (6), we obtain a system of differential Equations (16)–(21).

$$\begin{aligned} u_{1A} &= R_{1o}i_{1A} + L_{1o}\frac{di_{1A}}{dt} - \frac{M_m}{2}\left(\frac{di_{1B}}{dt} + \frac{di_{1C}}{dt}\right) + \\ &+ M_m\left(\frac{di_{2A}}{dt}\cos\alpha + \frac{di_{2B}}{dt}\cos\left(\alpha + \frac{2\pi}{3}\right) + \frac{di_{2C}}{dt}\cos\left(\alpha + \frac{4\pi}{3}\right)\right) - \\ &- \omega M_m\left(i_{2A}\sin\alpha + i_{2B}\sin\left(\alpha + \frac{2\pi}{3}\right) + i_{2C}\sin\left(\alpha + \frac{4\pi}{3}\right)\right); \end{aligned} \qquad (16)$$

$$\begin{aligned} u_{1B} &= R_{1o}i_{1B} + L_{1o}\frac{di_{1B}}{dt} - \frac{M_m}{2}\left(\frac{di_{1A}}{dt} + \frac{di_{1C}}{dt}\right) + \\ &+ M_m\left(\frac{di_{2A}}{dt}\cos\left(\alpha + \frac{4\pi}{3}\right) + \frac{di_{2B}}{dt}\cos\alpha + \frac{di_{2C}}{dt}\cos\left(\alpha + \frac{2\pi}{3}\right)\right) - \\ &- \omega M_m\left(i_{2A}\sin\left(\alpha + \frac{4\pi}{3}\right) + i_{2B}\sin\alpha + i_{2C}\sin\left(\alpha + \frac{2\pi}{3}\right)\right); \end{aligned} \qquad (17)$$

$$\begin{aligned} u_{1C} &= R_{1o}i_{1C} + L_{1o}\frac{di_{1C}}{dt} - \frac{M_m}{2}\left(\frac{di_{1A}}{dt} + \frac{di_{1B}}{dt}\right) + \\ &+ M_m\left(\frac{di_{2A}}{dt}\cos\left(\alpha + \frac{2\pi}{3}\right) + \frac{di_{2B}}{dt}\cos\left(\alpha + \frac{4\pi}{3}\right) + \frac{di_{2C}}{dt}\cos\alpha\right) - \\ &- \omega M_m\left(i_{2A}\sin\left(\alpha + \frac{2\pi}{3}\right) + i_{2B}\sin\left(\alpha + \frac{4\pi}{3}\right) + i_{2C}\sin\alpha\right); \end{aligned} \qquad (18)$$

$$\begin{aligned} 0 &= R_{2o}i_{2A} + L_{2o}\frac{di_{2A}}{dt} - \frac{M_m}{2}\left(\frac{di_{2B}}{dt} + \frac{di_{2C}}{dt}\right) + \\ &+ M_m\left(\frac{di_{1A}}{dt}\cos\alpha + \frac{di_{1B}}{dt}\cos\left(\alpha + \frac{4\pi}{3}\right) + \frac{di_{1C}}{dt}\cos\left(\alpha + \frac{2\pi}{3}\right)\right) - \\ &- \omega M_m\left(i_{1A}\sin\alpha + i_{1B}\sin\left(\alpha + \frac{4\pi}{3}\right) + i_{1C}\sin\left(\alpha + \frac{2\pi}{3}\right)\right); \end{aligned} \qquad (19)$$

$$\begin{aligned} 0 &= R_{2o}i_{2B} + L_{2o}\frac{di_{2B}}{dt} - \frac{M_m}{2}\left(\frac{di_{2A}}{dt} + \frac{di_{2C}}{dt}\right) + \\ &+ M_m\left(\frac{di_{1A}}{dt}\cos\left(\alpha + \frac{2\pi}{3}\right) + \frac{di_{1B}}{dt}\cos\alpha + \frac{di_{1C}}{dt}\cos\left(\alpha + \frac{4\pi}{3}\right)\right) - \\ &- \omega M_m\left(i_{1A}\sin\left(\alpha + \frac{2\pi}{3}\right) + i_{1B}\sin\alpha + i_{1C}\sin\left(\alpha + \frac{4\pi}{3}\right)\right); \end{aligned} \qquad (20)$$

$$\begin{aligned} 0 &= R_{2o}i_{2C} + L_{2o}\frac{di_{2C}}{dt} - \frac{M_m}{2}\left(\frac{di_{2A}}{dt} + \frac{di_{2B}}{dt}\right) + \\ &+ M_m\left(\frac{di_{1A}}{dt}\cos\left(\alpha + \frac{4\pi}{3}\right) + \frac{di_{1B}}{dt}\cos\left(\alpha + \frac{2\pi}{3}\right) + \frac{di_{1C}}{dt}\cos\right)\alpha - \\ &- \omega M_m\left(i_{1A}\sin\left(\alpha + \frac{4\pi}{3}\right) + i_{1B}\sin\left(\alpha + \frac{2\pi}{3}\right) + i_{1C}\sin\alpha\right). \end{aligned} \qquad (21)$$

We find the electromagnetic torque through the electromagnetic energy $W_{y'}$ concentrated in the air gap of the IM Equation (22):

$$M_{y'} = \frac{\partial W_{y'}}{\partial\alpha_{\widetilde{a}}} = \frac{1}{2}[i]o\frac{\partial[L]}{\partial\alpha_{\widetilde{a}}}, \qquad (22)$$

where $\alpha_{\widetilde{a}}$ is the geometric angle of the rotor rotation.

By multiplying the matrices included in Equation (22), we find an analytical expression of the electromagnetic moment of the three-phase IM Equation (23):

$$M_{y'} = p_i M_m \left[ \sin\left(\alpha + \tfrac{\pi}{3}\right)(i_{1A}i_{2C} + i_{1B}i_{2A} + i_{1C}i_{2B}) - \right.$$
$$\left. - \sin\alpha(i_{1A}i_{2A} + i_{1B}i_{2B} + i_{1C}i_{2C}) + \sin\left(\alpha - \tfrac{\pi}{3}\right)(i_{1A}i_{2B} + i_{1B}i_{2C} + i_{1C}i_{2A}) \right], \tag{23}$$

where $p_i$ is the pole pairs number.

The motion equation has the form (24)

$$J_\Sigma \frac{d\omega}{dt} = p_i \left( M_{y'} - M_c \right) \tag{24}$$

where $M_c$ is the static moment; $J_\Sigma$ is the total inertia moment of the moving parts.

Analysis of Equations (16)–(21) shows that they contain functions of the electric angle of rotation of the rotor $\alpha$, so the system of differential equations describing electromagnetic processes in a three-phase asynchronous motor is a system of equations with variable coefficients. The solution of such systems is possible only by numerical methods. The solution result of the system of Equations (16)–(21) is in the form of currents in the rotor and stator phases of the windings.

To solve the aforementioned system of differential Equations (16)–(21) numerically, it is necessary to reduce them to normal form. Such reduction is carried out by numerical method of integration of the system of differential Equations (16)–(21) by method of solution of system of linear algebraic equations relative to derivatives. In this case, the system of Equations (16)–(21) is transformed so that their left-hand sides contain derivatives of desired functions with appropriate coefficients, and their right-hand sides contain desired functions.

Another way to solve the system of linear algebraic equations with respect to derivatives is to find the inverse matrix $[L]^{-1}$. In this case equations of system (6) should be presented in the form (25)

$$\frac{d}{dt}\begin{bmatrix} [i_1] \\ [i_2] \end{bmatrix} = \begin{bmatrix} [L_1] & [M_{12}] \\ [M_{21}] & [L_2] \end{bmatrix}^{-1} \left\{ \begin{bmatrix} [u_1] \\ [u_2] \end{bmatrix} - \begin{bmatrix} [R_1] & [0] \\ [0] & [R_2] \end{bmatrix} \begin{bmatrix} [i_1] \\ [i_2] \end{bmatrix} - \omega \frac{\partial}{\partial\alpha} \begin{bmatrix} [L_1] & [M_{12}] \\ [M_{21}] & [L_2] \end{bmatrix} \begin{bmatrix} [i_1] \\ [i_2] \end{bmatrix} \right\}. \tag{25}$$

The matrix $[L]$ contains variable coefficients, so the inverse matrix $\left[L^{-1}\right]$ must be determined at each integration step.

The expression of the instantaneous power consumption (26):

$$p_1 = u_{1A}i_{1A} + u_{1B}i_{1B} + u_{1C}i_{1C}. \tag{26}$$

Thus, the system (16)–(21) together with the expression of electromagnetic moment (23), equation of motion (23) and Equation (13) represents a mathematical model of a three-phase squirrel-cage induction electric motor in phase axes, without considering saturation of magnetic wire and losses in steel. The advantage of the considered mathematical description of electromechanical energy conversion processes in three-phase IM is that it uses instantaneous values of currents and voltages of stator and rotor winding phases as variables. This description of the dynamics gives a direct indication of the physical processes in the motor.

Control in the coordinate system oriented by the calculated rotor magnetic flux vector $(\psi_R)$ is called direct-field oriented control. Usually, systems with direct orientation also contain a flow stabilization loop.

But a simpler approach and another class of systems have emerged in which the rotor magnetic flux vector $(\psi_R)$ is not evaluated, but its angle is calculated from the slip and velocity (or position) of the rotor. Such systems are called systems with indirect field orientation.

Such systems do not contain a flow stabilization loop. In this case, it is believed that if the control actions on the motor are formed based on the condition of flow constancy, then if they are accurately implemented, the flow will be constant even without stabilization.

As practice has shown, such an approach has quite justified itself due to a good ratio of accuracy and simplicity of regulation.

When operating at rated voltage with increasing frequency, the amplitude of the stator electric current is limited by the increasing inductive resistance of the IM. Therefore, as the frequency increases, the maximum achievable motor torque is also limited.

For controller synthesis and simulation, a dynamic model of the IM developed in the system (*d-q*) that is synchronized with the flux of the stator was created. The nonlinear IM model can be expressed as follows

$$\frac{dI_{sd}}{dt} = a_1 I_{sd} + a_2 \hat{\psi}_r + \omega I_{sq} + b_1 \frac{\left(I_{sq}\right)^2}{\hat{\psi}_r} + cV_{sd}$$

$$\frac{dI_{sq}}{dt} = a_1 I_{sq} - a_3 \omega \hat{\psi}_r - \omega I_{sd} - b_1 \frac{I_{sd} I_{sq}}{\hat{\psi}_r} + cV_{sq}$$

$$\frac{d\hat{\psi}_r}{dt} = b_2 \omega \hat{\psi}_r + b_1 I_{sd}$$

$$\frac{d\omega}{dt} = \frac{1}{J} \left(m\hat{\psi}_r I_{sq} - T_L - f\omega\right)$$

$$\frac{d\theta_s}{dt} = \omega + b_1 \frac{I_{sd} I_{sq}}{\hat{\psi}_r}$$

where $I_{sd}$ and $I_{sq}$, $V_{sd}$ and $V_{sq}$ are constant and quadratic components of the stator electric current and voltage vector, respectively; $\hat{\psi}_r$ is the rotor magnetic flux, which is estimated, $N_p$ is the number of pole pairs, $\omega$ is speed of the rotor, $f$ is the viscous friction; $J$ is the inertia moment; $T_L$ is load torque, and $\theta_s$ is angle of the stator field.

As mentioned in the literature [29], expressions $a_1$, $a_2$, $a_3$, $b_1$, $b_2$, $c$, and $m$ depend on parameters of IM as follows

$$a_1 = \left(\frac{R_s}{\sigma L_s} + \frac{M_{sr}^2}{\sigma L_s L_r T_s}\right)$$

$$a_2 = \frac{M_{sr}}{\sigma L_s L_r T_s}$$

$$a_3 = \frac{M_{sr}}{\sigma L_s L_r}$$

$$c = \frac{1}{\sigma L_s}$$

$$\sigma = 1 - \frac{M_{sr}^2}{L_s L_r}$$

$$T_s = \frac{L_s}{R_s}$$

$$T_r = \frac{L_r}{R_r}$$

$$b_1 = \frac{M_{sr}}{T_r}$$

$$b_2 = -\frac{1}{T_r}$$

$$m = \frac{N_p M_{sr}}{L_r}$$

where, $R_r$ is the rotor resistance and $R_s$ is the stator resistance;

$L_s$ is magnetization inductance of stator and $L_r$ is magnetization inductance of rotor;

$T_s$ and $T_r$ are constants for stator and rotor, respectively;

$M_{sr}$ is stator and rotor mutual inductance.

The electromagnetic torque expressed as a function of the stator current $I_{sq}$ and the $\hat{\psi}_r$ rotor magnetic flux is shown below:

$$T_e = \frac{3}{2}\frac{N_p M_{sr}}{L_r}\hat{\psi}_r I_{sq}$$

The IM is powered by a PWM inverter. The motor propulses a mechanic load characterized by inertia *J*, load torque $T_L$, and viscous friction coefficient *f*. A PI regulator is used for the speed control loop to obtain an electric current set point $i_q$ which regulates torque of the IM. Magnetic flux is driven by the $i_d$ axis current set point. The IM speed, electric current, and torque signals are available in the MatLab/Simulink software at the IM block output (Figure 1). A simulation model of IM is developed in MatLab/Simulink software.

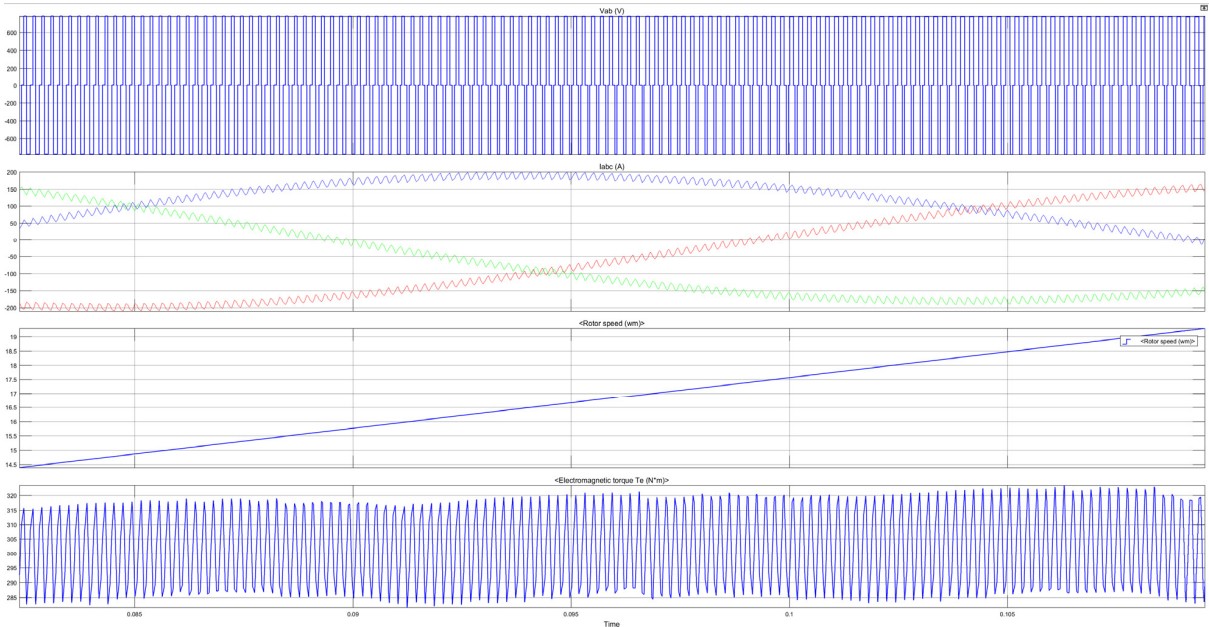

**Figure 1.** Diagrams of voltage changes between two phases, electric current in three phases, angular speed of motor and torque.

Model implements a three-phase asynchronous machine (wound rotor, squirrel cage or double squirrel cage) modeled in a selectable *dq* reference frame (rotor, stator, or synchronous). Stator and rotor windings are connected in wye to an internal neutral point. IM parameters are shown in Table 1.

**Table 1.** Parameters of the induction motor.

| Parameters | Value |
|---|---|
| Nominal power, voltage (line-line), and frequency [$P_n$(VA), $V_n$(Vrms), $f_n$(Hz)] | $50 \times 746$, 460, 60 |
| Stator resistance and inductance [$R_s$(Ohm), $L_{ls}$(H)] | $0.087, 0.8 \times 10^{-3}$ |
| Rotor resistance and inductance [$R_{r'}$(Ohm), $L_{lr'}$(H)] | $0.228, 0.8 \times 10^{-3}$ |
| Mutual inductance [$L_m$ (H)] | $34.7 \times 10^{-3}$ |
| Inertia, friction factor, pole pairs [$J$(kg· m$^2$), $F$(N· m· s), p(-)] | 1.662, 0.1, 2 |

## 3. Results

The results of the IM simulation and comments on these graphs are presented below. Figure 1 shows graphs of the voltage change between two phases, the electric current in three phases, the angular speed of motor, and torque.

At the final part of the simulation corresponding to time of 2–3 s, the stationary state of the system has been reached, as presented in Figure 2.

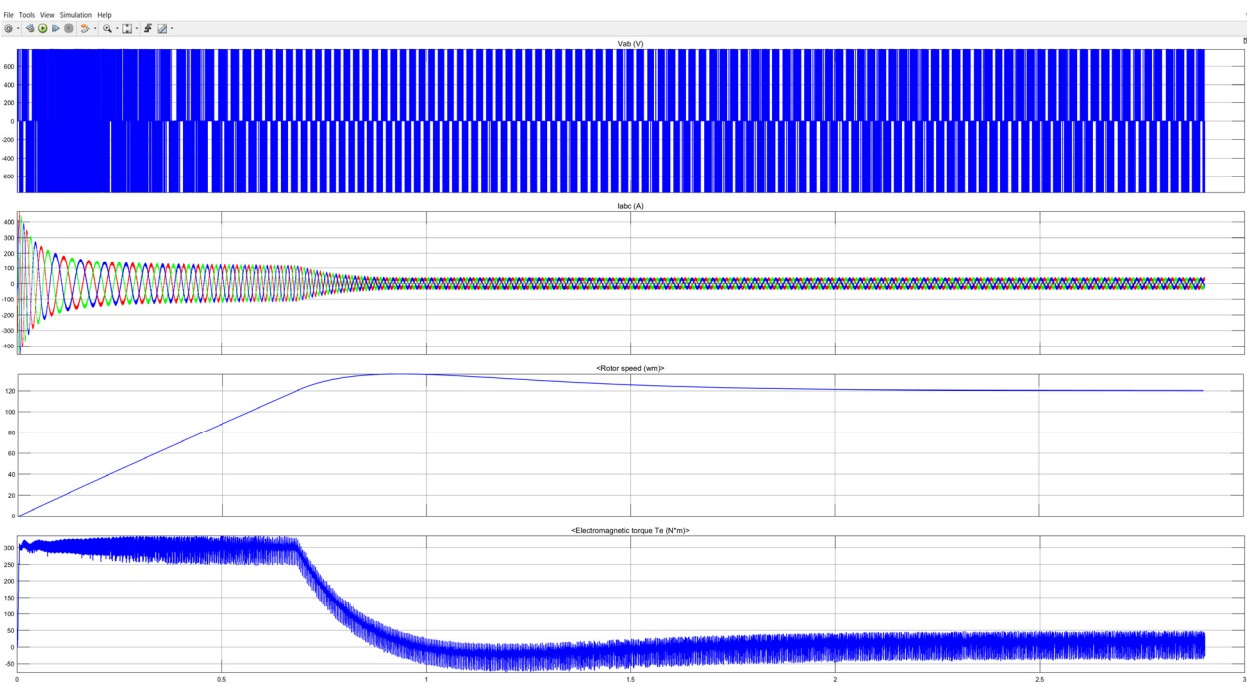

**Figure 2.** Diagrams of voltage changes between two phases, electric current in three phases, angular speed of the motor and torque.

The following part describes the system behaviour in reaction to changes in the set values of load torque and speed. The state vector of the initial conditions "xInitial" for a run corresponding to $w_m$ = 120 rad/s and $T_L$ = 0 N·m is loaded into the workspace immediately after the start of the simulation (Model Properties). When using two Manual Switch blocks switching from the blocks "Constant Speed" and "Constant Torque" to Step blocks (the set speed $w_{ref}$ at time of 0.2 s is ranging from 120 to 160 rad/s, and $T_L$ at time of 1.8 s varies from 0 to 200 N·m). The response of the actuator to load torque and speed variations is presented in Figure 3. Figure 4 shows the model of IM control with measurement of signals at the gates of IGBT transistors generating PWM.

Based on the model-oriented design of the control program, the microprocessor unit that implements IM vector control and the electronic unit that generates PWM signals is formed [30]. The design procedure for the LAUNCHXL-F28379D controller and DRV8312-69M-KIT inverter is shown in the Motor Control Blockset software.

After selecting Build, Deploy & Start placed at the Hardware tab of the Simulink model, the C code, CCS project, and target .out file are generated. A serial link is utilized to transfer .out the file target from the computer to the microprocessor and to run the downloaded algorithm.

Update of the configuration parameters of the Simulink-created model should be performed prior to the simulation start. Another option is to upload the model into the microcontroller. In the Simulink window, Hardware Settings can be selected from the Hardware tab that activates the dialog box of the Configuration Parameters. After that, the target computer selecting follows in the Hardware board menu, shown in Figure 5.

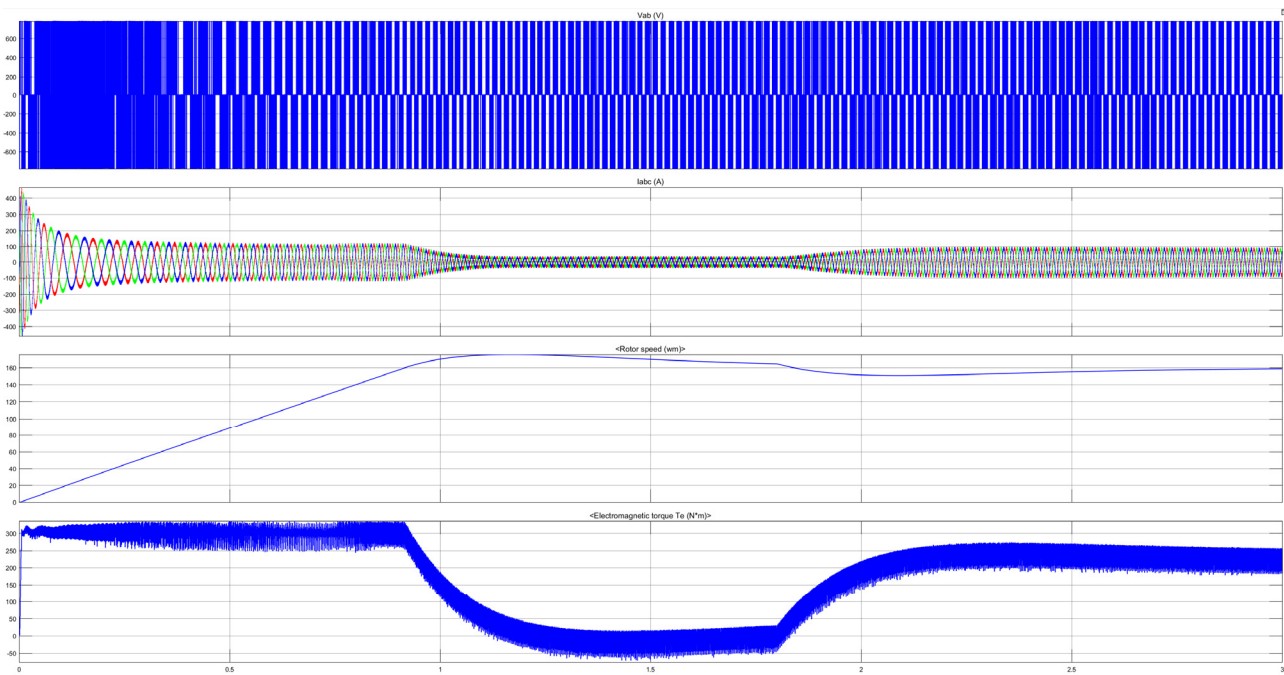

**Figure 3.** Diagrams of voltage changes between two phases, electric current in three phases, angular speed of motor and torque with the change of speed and load.

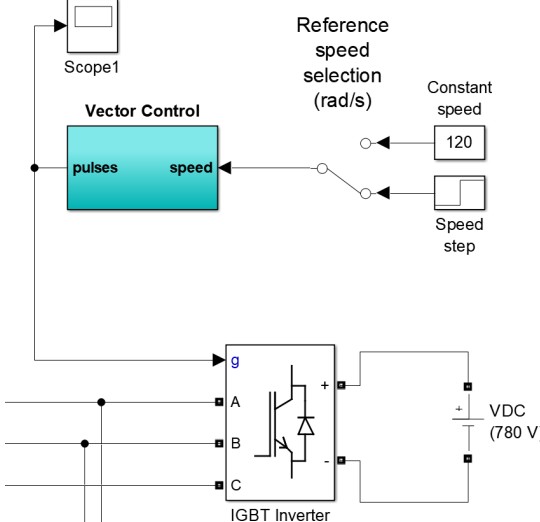

**Figure 4.** Model of IM control with measurement of signals on IGBT transistors gates forming PWM.

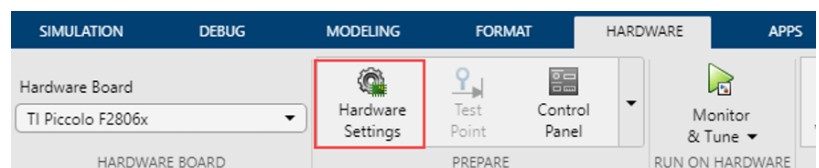

**Figure 5.** Simulink Hardware configuration window.

To configure the Solver at the eponymous tab of the dialog box of Configuration Parameters menu, the fixed step of the discrete solver should be selected and the "auto" option for the Fixed-step size [31]. Figure 6 presents the corresponding Configuration Parameters window.

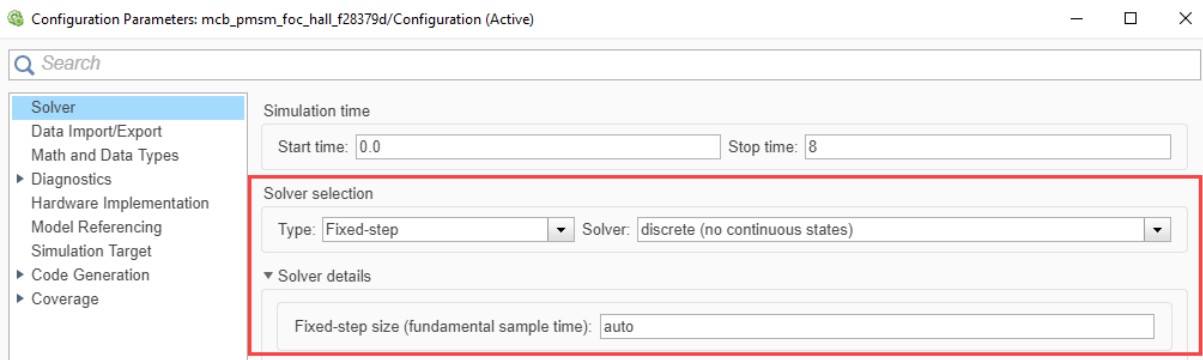

**Figure 6.** Configuration Parameters window.

If a shaft speed feedback encoder is to be used, the encoder is to be connected with the microcontroller board and the dialog box parameters in the Configuration Parameters are to be configured as follows:

1. Hardware Implementation tab must be opened.
2. Hardware board settings must be selected, then Target hardware resources and eQEP group.

The example in Figure 7 presents the eQEP setup for an example of quadrature encoder connected to the board LAUNCHXL-F28379D.

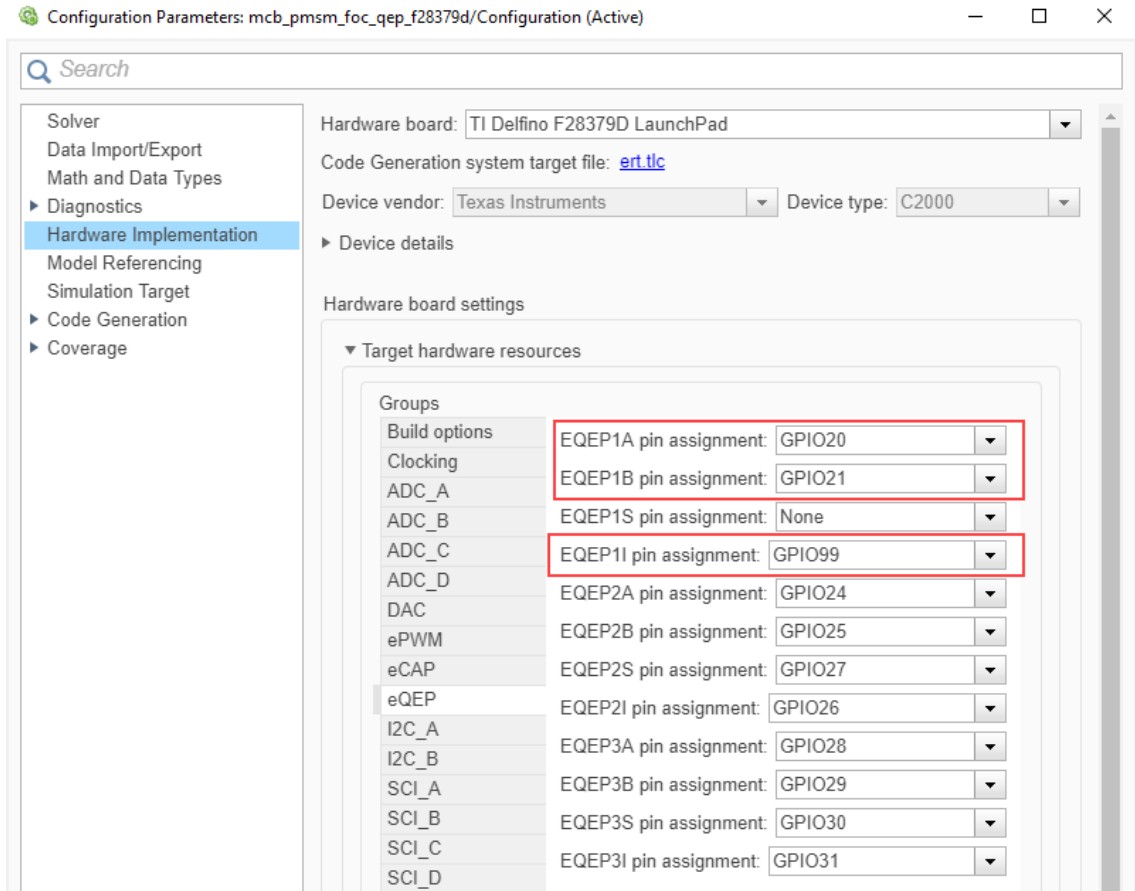

**Figure 7.** Setup for quadrature encoder connected to LAUNCHXL-F28379D board.

## 4. Discussion

A mathematical model of IM has been developed using the (*d-q*) rotating system of co-dynamics that was constructed. Then, an IM model was developed in the state space. The proposal of IM control was also designed on the base of model-oriented approach. The simulation was performed in the MatLab/Simulink program [32,33]. The LAUNCHXL-F28379D board was used as a microprocessor-based control system. With this approach it is possible to quickly design microprocessor control systems with a minimum of errors. Also, it is possible to conduct semi-natural experiments, when the control of IM is carried out according to its model set in MatLab/Simulink program.

The next continuation of our research will be oriented to the design of intelligent motor drive systems, which should be able to detect defects in the motor itself and in the power transistors [34,35].

For fault detection, tools for so-called progressive diagnostics of electrical drives in robots and machine tools are proposed with the support of sensors [36], computational systems and artificial intelligence methods based on neural networks [37], and the development of control algorithms based on discrete models [38]. We consider the use of data analysis methods and process data for the purpose of fault prediction and process optimization [39,40], and data mining methods [41]. Similar to [42], the diagnosis of electric actuators based on induction machines is addressed using an artificial intelligence model containing a description of the source of linguistic variables and technical conditions of a database system of fuzzy inference rules to identify the technical condition of electric actuators. The next step of the research will be to develop a fault-tolerant intelligent motor control based on knowledge of defects.

## 5. Conclusions

In this paper, a mathematical simulation model of the IM is described. More details of an IM dynamical model in the rotating (*d-q*) coordinate system in the state space that is synchronized with the stator flux is described. In addition, the IM simulation model is developed in the MatLab/Simulink program with the possibility of setting the angular velocity of rotation and moment of resistance. Moreover, an example of hardware configuration comprising the model-oriented design of the control program on the example of digital signal processors from Texas Instruments on the LAUNCHXL-F28379D board is presented.

The results of the simulation of the motor when the speed of rotation is increased and the speed is further stabilized, as well as during the increasing load, are obtained. The operation of the motor is controlled using the PWM regulator based on IGBT. It is obvious from the results that the parameters of the inverter are set correctly. The electric current flowing through the motor phases has a sinusoidal form. The method of a model-oriented design with a selection of a particular microcontroller and encoder for speed feedback is also shown.

The application of such a methodology of program design for control systems has good potential to reduce the IM development time and the number of programming errors, which proves that this tool is prospective for use in highly competitive sectors for applications e.g., in the automation environment.

**Author Contributions:** Conceptualization, T.K., Y.N. and P.B.; methodology, Y.N. and P.B.; software, Y.N.; validation, T.K., Y.N. and P.B.; formal analysis, T.K., Y.N. and P.B.; investigation, T.K., Y.N. and P.B.; resources, Y.N.; data curation, T.K., Y.N. and P.B.; writing—original draft preparation, Y.N. and P.B.; writing—review and editing, T.K. and Y.N.; visualization, T.K. and Y.N.; supervision, T.K. and Y.N.; project administration, T.K., Y.N. and P.B.; funding acquisition, T.K. All authors have read and agreed to the published version of the manuscript.

**Funding:** This work was funded by the Slovak Ministry of Education within project VEGA No. 1/0823/21 and by the Slovak Research and Development Agency under contract No. APVV-18-0316.

**Institutional Review Board Statement:** Not applicable.

**Informed Consent Statement:** Not applicable.

**Conflicts of Interest:** The authors declare no conflict of interest. The funders had no role in the design of the study; in the collection, analyses, or interpretation of data; in the writing of the manuscript, or in the decision to publish the results.

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
