# Peer review of "Model-Based Design of Induction Motor Control System in MATLAB"

_applsci, doi:10.3390/app122311957_

Round 1
Reviewer 1 Report
The paper should be accepted after addressing the following minor comments. page 3 - line 2: add 'steady' just before the word 'mode' page-5 - line 2 after Eqn. (1)
change definition of phi to 'flux vector' - current definition 'vector of currents' is confusing.
Author Response
Thank you so much for your detailed reading and recommendations provided in the review. Your valuable comments helped us to improve the quality of our article.
Now, the proposed corrections are implemented in the article.
Reviewer 2 Report
1- The Novelty, objective and purpose of the research are not clearly defined.
2- Key findings are not elaborated in the abstract and conclusion.
3- Any comparison with existing models is not established.
4- Overall structure of the article is not structured considering readers' perspectives.
5- Overall, the problem statement is not defined. The authors need to clearly define the purpose of the research.
6- The article seriously needs to be overhauled in terms of the information provided, structure, key findings, any comparison with existing models, what are the advantages of the presented model.
Author Response
Thank you so much for the review and valuable comments aimed at improving the quality of our article. For more details, please see the attached file and the manuscript.

Round 2
Reviewer 2 Report
The authors have incorporated the comments highlighted.